# Combined Dopamine and Grape Seed Extract-Loaded Solid Lipid Nanoparticles: Nasal Mucosa Permeation, and Uptake by Olfactory Ensheathing Cells and Neuronal SH-SY5Y Cells [note 1]

**DOI:** 10.3390/pharmaceutics15030881

**Published:** 2023-03-08

**Authors:** Adriana Trapani, Stefano Castellani, Lorenzo Guerra, Elvira De Giglio, Giuseppe Fracchiolla, Filomena Corbo, Nicola Cioffi, Giuseppe Passantino, Maria Luana Poeta, Pasqualina Montemurro, Rosanna Mallamaci, Rosa Angela Cardone, Massimo Conese

**Affiliations:** 1Department of Pharmacy-Drug Sciences, University of Bari “Aldo Moro”, 70125 Bari, Italy; 2Department of Precision and Regenerative Medicine and Ionian Area (DiMePRe-J), University of Bari “Aldo Moro”, 70125 Bari, Italy; 3Department of Biosciences, Biotechnologies and Environment, University of Bari “Aldo Moro”, 70125 Bari, Italy; 4Department of Chemistry, University of Bari “Aldo Moro”, 70125 Bari, Italy; 5Department of Veterinary Medicine, Pathological Anatomy, University of Bari “Aldo Moro”, 70125 Bari, Italy; 6Department of Clinical and Experimental Medicine, University of Foggia, 71122 Foggia, Italy

**Keywords:** colloids, dopamine, grape seed extract, intranasal administration, olfactory ensheathing cells, SH-SY5Y cells

## Abstract

We have already formulated solid lipid nanoparticles (SLNs) in which the combination of the neurotransmitter dopamine (DA) and the antioxidant grape-seed-derived proanthocyanidins (grape seed extract, GSE) was supposed to be favorable for Parkinson’s disease (PD) treatment. In fact, GSE supply would reduce the PD-related oxidative stress in a synergic effect with DA. Herein, two different methods of DA/GSE loading were studied, namely, coadministration in the aqueous phase of DA and GSE, and the other approach consisting of a physical adsorption of GSE onto preformed DA containing SLNs. Mean diameter of DA coencapsulating GSE SLNs was 187 ± 4 nm vs. 287 ± 15 nm of GSE adsorbing DA-SLNs. TEM microphotographs evidenced low-contrast spheroidal particles, irrespective of the SLN type. Moreover, Franz diffusion cell experiments confirmed the permeation of DA from both SLNs through the porcine nasal mucosa. Furthermore, fluorescent SLNs also underwent cell-uptake studies by using flow cytometry in olfactory ensheathing cells and neuronal SH-SY5Y cells, evidencing higher uptake when GSE was coencapsulated rather than adsorbed onto the particles.

## 1. Introduction

In western countries, Parkinson’s disease (PD) is a common movement disorder in over-60-year-old patients. Extrapyramidal motor symptoms depend on the dopaminergic pathway degeneration in the patient’s brain, where reduced levels of the neurotransmitter dopamine (DA) are found in the Substantia Nigra. Although clinical diagnosis is mainly established on the base of bradykinesia, PD is related to several nonmotor symptoms that are responsible for overall disability. Among factors contributing to the development of this pathology, genetic and environment-related causes are the major ones liable for PD and, as a not-less-important factor, reactive oxygen species (ROS) amounts are to be mentioned because they significantly contribute to neurodegenerative age-related disorders [1].

Precisely, oxidative stress associated to PD in aging people is involved in the death of dopaminergic neurons and, hence, reduced systemic antioxidant capacity has been identified as a biomarker of PD. Moreover, such compounds have also been correlated with nonmotor abnormalities (i.e., olfactory impairment and dysautonomia) [2].

From a clinical viewpoint, however, once diagnosis of PD is claimed, then levodopa (L-Dopa) treatment is started in order to supply the missing neurotransmitter DA, it being a well-known bioprecursor of DA. Indeed, DA does not overcome the blood–brain barrier (BBB) because of its physicochemical features and extensive metabolism when orally administered [3]. In the case of L-Dopa, an active transport system allows it to bypass the BBB, and the enzyme L-Dopa-decarboxylase in the brain converts it into DA by a decarboxylation reaction [3,4]. Although L-Dopa still represents the reference drug for PD treatment, when it is recommended at a high dosage for long-term treatment, then its metabolic reactions are well-known to contribute to PD progression [5].

To prevent high levels of ROS species, which negatively influence the PD patients’ mental health, L-Dopa chronic therapy should be matched with additional application of free radical scavengers and methyl group bearing vitamins [5].

Hence, such therapeutic protocols point out that new perspectives of PD treatment can be disclosed, namely, to identify appropriate strategies for codelivery of synergic agents involved in PD treatment. Currently, a notable technological innovation in the field of medicine in terms of exploring carrier systems, whose aim is the simultaneous delivery of two active principles to the targeted site, has been made in this respect. In the case of PD treatment, according to the approach of modified drug delivery systems, the combination of a dopaminergic drug with antioxidant agent(s) can be achieved, for instance, by the use of liposomes, solid lipid nanoparticles (SLNs), nanoemulsions, self-emulsifying drug delivery systems or niosomes. All mentioned systems are among the novel drug delivery system carriers already studied for enhancing the pharmacological supply to PD patients and to solve biopharmaceutical issues, including poor bioavailability and instability [6,7].

Among the above-mentioned carriers, when SLNs are produced, similarly to that occurring also with liposomes, they offer the benefit of loading not only hydrophobic drug substances but also hydrophilic ones, thanks to the presence of a lipid-forming matrix and a polar surfactant stabilizing the resulting nanocarrier, respectively. Additionally, they can also allow sustained release together with safety and efficacy in different pathological conditions [8,9,10].

In our previous works, we have already exploited the potential of SLNs for the nose-to-brain (NTB) delivery of DA [11,12]. Moreover, recently, in the literature, the release of DA in the brain has been seen to be enhanced after the administration of an antioxidant agent together with a DA agonist via SLN carrier, thus reinforcing the role exerted by this type of adopted nanocarrier together with the selection of the antioxidant compound to be codelivered [13].

However, the BBB is a great hurdle for brain drug delivery and, consequently, how to approach the central nervous system (CNS) pathology treatment is still a challenge. The combination of nanocarriers cited above, with their capacities to noninvasively overcome the BBB for the treatment of CNS diseases [14,15], and the NTB way of delivering therapeutic molecules, offer unprecedented modalities to achieve this task. In particular, NTB relies on different pathways, among which is the uptake of nanocarriers by the olfactory epithelium and their interaction with the nerve endings of olfactory receptor neurons. This would allow passage to the olfactory bulb, following the nerve channel due to the olfactory ensheathing cells [16].

Moving from this state of art, in this work, we aimed at investigating a novel combination of DA and Apulian grape seed extract (GSE) in an SLN carrier, with the purpose to gain more insights into the use of this combination of bioactive substances, as well as to achieve a colloidal vehicle suitable for NTB for PD applications. The relevance of GSE as an effective antioxidant mixture was shown in some recent papers, where the enhanced antioxidant and antibacterial properties of such a natural mixture with possible application in food packaging [17], as well the oxidative stress decrease in respiratory syndromes [18], were described. The obtained SLNs combining DA and GSE were herein evaluated for their physicochemical properties, cytobiocompatibility and cellular uptake and were also compared with DA-loaded SLNs physically adsorbing GSE previously described by us [19].

## 2. Materials and Methods

### 2.1. Materials

Grape seed extract containing ≥95.0% of proanthocyanidins was a gift from Farmalabor (Canosa di Puglia, Italy), whereas Gelucire^®®^ 50/13 was provided by Gattefossé (Milan, Italy). Dopamine hydrochloride, carboxyl ester hydrolase (E.C. 3.1.1.1, 15 units/mg powder), Fluorescein 5(6)-isothiocyanate (FITC) and Tween^®®^ 85, as well as the salts used for buffer preparation, were bought from Sigma-Aldrich (Milan, Italy). In this work, double-distilled water was used, and all other chemicals were of reagent grade. Dopamine–fluorescein isothiocyanate (DA-FITC) was synthesized by employing DA and FITC, as described previously [20].

### 2.2. Preparation of SLNs

#### 2.2.1. DA-SLNs Coencapsulating GSE

DA-loaded Gelucire^®®^ 50/13 SLNs were prepared following the melt homogenization method [21]. Gelucire^®®^ 50/13 (60 mg) was melted at 70 °C. An amount of 6 mg of GSE was dispersed in the aqueous phase made of surfactant (Tween^®®^ 85, 60 mg) and 1.37 mL diluted acetic acid (0.01%, *w*/*v*) in a separate vial under homogenization at 12,300 rpm with an Ultra-Turrax model T25 apparatus (Janke and Kunkel, IKA^®®^-Werke GmbH and Co., Staufen Germany) and let to equilibrate for 30 min at 70 °C. Then, 10 mg of DA was introduced in the aqueous phase, the resulting mixture was added to the melted phase at 70 °C, and the emulsion was homogenized at 12,300 rpm for 2 min by Ultra-Turrax system. Then, the nanosuspension was cooled at room temperature to achieve DA-SLNs coencapsulating GSE. Such SLNs were centrifuged (16,000× *g*, 45 min, Eppendorf 5415D, Hamburg, Germany), and the pellet was harvested and resuspended in distilled water for further studies. Throughout the manuscript, the resulting SLNs were abbreviated “DA-co-GSE-SLNs”.

#### 2.2.2. GSE-Adsorbing DA-SLNs

For DA-SLNs adsorbing GSE, the first step of the method consisted of the preparation of DA-loaded SLNs, as reported elsewhere [21], but starting from 20 mg of DA rather than 10 mg to force initial DA *cargo*. After cooling down at room temperature, an aliquot of 0.5 mL of the resulting DA-SLNs was incubated with 1 mL of GSE aqueous solution (1 mg/mL concentration) at room temperature for 3 h in the dark, maintaining mild stirring (50 oscillations/min). Then, the mixture was centrifuged at 16,000× *g* for 45 min (Eppendorf 5415D), and the pellet was dispersed in distilled water. Throughout the manuscript, the resulting SLNs were abbreviated “GSE-ads-DA-SLNs”.

#### 2.2.3. Fluorescent SLNs

Two types of fluorescent carriers were formulated in view of the uptake experiments.

(a)FITC physically loaded. According to our previous work [21], instead of unlabeled DA, 10 mg of FITC in the aqueous phase was dispersed in order to achieve fluorescent SLNs in the absence of GSE, herein indicated as “FITC-SLNs”. Gelucire^®®^ 50/13, Tween 85 and diluted acetic acid were used as reported in Section 2.2.(b)FITC covalently bound to DA. Via the use of 10 mg of DA-FITC, fluorescent SLNs were obtained, in agreement with the procedure described in Section 2.2 and, in this case, they resulted to be “DA-FITC-SLNs co-encapsulating GSE” (DA-FITC-co-GSE-SLNs). Another fluorescent formulation of SLNs was produced, by adsorbing 1 mL of GSE aqueous solution (1 mg/mL) onto 0.5 mL of preformed DA-FITC-SLNs. The physical adsorption was performed for 3 h at room temperature and under light protection. The resulting SLNs were called “GSE-ads-DA-FITC SLNs”.

For all types of SLNs under investigation, plain SLNs (pSLNs), obtained as in Section 2.2 but in the absence of both DA and GSE, were taken as control [22].

### 2.3. Quantitative Determination of DA, GSE and FITC

The DA and GSE quantifications were performed by HPLC, as previously reported [23]. The HPLC apparatus included a Waters Model 600 pump (Waters Corp., Milford, MA, USA), a Waters 2996 photodiode array detector and a 20 μL loop injection autosampler (Waters 717 plus). A Synergy Hydro-RP (25 cm × 4.6 mm, 4 μm particles; Phenomenex, Torrance, CA) was the stationary phase, and a 0.02 M potassium phosphate buffer, pH 2.8: CH_3_OH 70:30 (*v*:*v*), was adopted as mobile phase. The isocratic mode was selected for column elution at the flow rate of 0.7 mL/min and, under such chromatographic conditions, the retention times of DA and GSE were found to be equal to 5.5 min and 12 min, respectively [19].

To determine DA and GSE content in both labeled and unlabeled SLNs, after freeze-drying (T = −50 °C; P = 0.1 mbar; Lio Pascal 5 P, Milan, Italy), particles were cleaved upon enzymatic digestion by esterases. The enzyme was dissolved at 12 I.U./mL in phosphate buffer (pH 5) and 1–2 mg of freeze-dried SLNs were incubated with 1 mL of the enzyme solution for 30 min in an agitated (40 rpm/min) water bath set at 37 °C (Julabo, Milan, Italy). Then, the resulting mixture was centrifuged (16,000× *g*, 45 min, Eppendorf 5415D) and the obtained supernatant was subjected to HPLC analysis, as above.

The fluorescence intensity of the FITC in supernatants of fluorescent SLNs was determined at a FITC concentration equal to 2 μg/mL after diluting it in 0.1 M of phosphate buffer, pH 8.0. For standardization, the calibration of the fluorometer (Perkin Elmer, Milan, Italy) employed solutions in the range from 1 to 140 ng/mL of FITC in phosphate buffer, pH 8.0, obtained from dilution in such a buffer of a previously prepared stock solution of 100 μg/mL FITC in methanol. For fluorometric analysis, excitation and emission wavelengths of 488 and 525 nm, respectively, were selected, and slits were set at 2.5 cm.

The encapsulation efficiency (E.E.%) was calculated by Equation (1):E.E.% = DA (GSE or FITC) in the supernatant after esterase assay/Total DA (GSE or FITC) × 100(1)
where total DA (GSE or FITC) is intended as the starting amount of each substance used for SLN preparation. The study was performed in triplicate.

### 2.4. Physicochemical Characterization of SLNs

For all SLNs, particle size and polydispersity index (PDI) determinations at 25 °C took place after dilution in double-distilled water (1:1, *v*:*v*) in disposable polystyrene latex cuvettes with angle detection set at 90°. A Zetasizer NanoZS (ZEN 3600, Malvern, UK) equipment was used following photon correlation spectroscopy (PCS) mode. For data accumulation, eleven runs of counting were used, and each run was 10 s. By using laser Doppler anemometry (Zetasizer NanoZS, ZEN 3600, Malvern, UK) following dilution 1:20 (*v*:*v*) in the presence of KCl (1 mM, pH 7), zeta potential measurements were also carried out at 25 °C in disposable polystyrene latex folded cuvettes after 120 s of equilibration time and performing three measurements, each one having 10–100 runs. The particle size, PDI and zeta potential values were each measured in eight sample replicates [24]. Transmission electron microscopy (TEM, FEI Tecnai 12 TEM, Eindhoven, The Netherlands), equipped with a LaB6 filament operating at 120 kV, was adopted to investigate the nanoparticle morphology, at the dried state. For SLN observations, drops of suspensions were deposited on a Formvar^®®^-coated Cu grid (300 mesh, Agar Scientific, Stansted, UK). The microscope was calibrated using the S106 Cross Grating (2160 lines/mm, 3.05 mm) supplied by Agar Scientific. Corrections concerning alignments and astigmatism were carried out following factory settings and fast Fourier-transform processing, respectively.

### 2.5. Solid State Studies

#### 2.5.1. Fourier-Transform Infrared (FT-IR) Spectroscopy

FT-IR spectra were acquired in KBr discs using 2–5 mg of pure DA, GSE and lyophilized particles obtained after 72 h of a freeze-drying cycle (Lio Pascal 5P, Milan, Italy). Tested samples included plain SLNs, DA-co-GSE SLNs and GSE-ads-DA-SLNs. A Perkin Elmer 1600 FT-IR spectrometer (Perkin Elmer, Milan, Italy) processed all the spectra (r.t., 4000–400 cm^−1^ wavenumber range) at a resolution of 1 cm^−1^ [25,26].

#### 2.5.2. Thermal Analysis via Differential Scanning Calorimetry (DSC)

DSC calorimetric runs were carried out for bulk materials, freeze-dried DA-co-GSE SLNs and freeze-dried GSE-ads-DA-SLNs using a Mettler Toledo DSC 822e STARe 202 system combined with a DSC MettlerSTARe software v.2 (Mettler Toledo, Milan, Italy). About 5 mg of each product was placed in an aluminum pan and hermetically sealed. The scanning rate was 5 °C/min under a nitrogen flow of 20 cm^3^/min, and the temperature range was set from 25 to 275 °C for all samples. Following the procedure of the MettlerSTARe Software, the DSC apparatus was calibrated using indium (99.9%). Each thermal run was replicated three times.

### 2.6. Assessment of Physical Stability of SLNs

To evaluate GSE-ads-DA-SLNs and DA-co-GSE-SLNs physical stability, measurements of their particle dimensions were carried out, storing at 4 °C in the refrigerator up to 3 months, as well as at 25 °C and 37 °C for up to one week and 24 h, respectively. Precisely, a water bath (Julabo, Milan, Italy), set at the temperatures of 25 °C and 37 °C, with agitation of 40 rpm/min, allowed SLN incubation for the above-mentioned time frames. At different time points, particle size was acquired according to the analysis described in Section 2.4. For each type of SLN, the assay was performed in triplicate at each temperature.

### 2.7. Ex Vivo SLN Permeation Studies

By the use of a vertical Franz cell (PermeGear Inc., SES GmbH, Bechenheim, Germany [27,28]), ex vivo permeation studies were carried out through excised nasal porcine mucosa. The ethics committee of the University of Bari (Italy) approved the study protocol. The pig snouts were obtained from a local slaughterhouse, and within 10 min after killing the animals, the samples were stored in dry ice. Before their use, in order to exclude pathologies and any mucosal damage, a portion of the nasal mucosa was immediately stored in 10% buffered formalin and the histological observation was performed for each animal at least 72 h after fixation. The use of paraffin allowed tissue embedding; sections were cut into serial sections 5 μm thick, and hematoxylin and eosin or periodic acid-Schiff were used for staining and were examined with a Leica DMLS D 4000 microscope equipped with digital camera (Leica DMC5400, Milan, Italy). Sections were rated by working with magnification of ×200 (i.e., objective × 20 and eyepiece lens × 10; 0.7386 mm^2^ per field), subsequent to an ×400 × field (i.e., objective × 400 and eyepiece lens ×10; 0.1885 mm^2^ per field).

For the experiments, the anterior surface of the mucosa was faced towards the donor compartment (effective diffusion area = 0.6 cm^2^), and freeze-dried DA-co-GSE-SLNs (or GSE-ads-DA-SLNs) were resuspended in 100 μL of simulated nasal fluid (SNF, pH 5–6 [29,30]) and placed in the donor compartment. The Franz cell apparatus was maintained at 37.00 ± 0.03 °C with a CD-B5 heating circulator bath (Julabo GmbH, Seelbach, Germany) for 24 h, and the acceptor compartment was filled with 5 mL of SNF in the presence of an ATE magnetic stirrer (VELP Scientifica Srl, Usmate, Italy) that allowed continuous stirring. The assembled system was allowed to equilibrate at 37 °C for 15 min, and isothermal conditions were maintained through a water jacket system. At predetermined time points (0, 1, 2, 4, 8 and 24 h), aliquots of 0.5 mL were withdrawn from the receiving chamber and refilled with 0.5 mL of fresh SNF. Each withdrawal was subjected to centrifugation (Eppendorf 5415D, Hamburg, Germany) at 13,200× *g* for 45 min, to allow the discarding of the pellet of some very small SLN particles eventually diffusing as such. After centrifugation, the resulting supernatant was analyzed according to HPLC method described in Section 2.3. Controls were represented by the physical mixtures of the powders of 10 mg of DA and 6 mg of GSE placed, as above, in the donor compartment after resuspension in SNF. For each formulation, the permeation study was repeated in sextuplicate.

The apparent permeability coefficient (*P*_app_) was obtained according to the Equation (2):
*P*_app_ = dQ/dtAC_0_
(2)
where dQ/dt is the cumulative amount of DA permeated vs. time per unit of area obtained by the slope of the linear tract of the first segment of the curve, A is the effective surface area, and C_0_ is the initial DA concentration in the donor compartment [31,32]. When the experiment was concluded, the nasal porcine mucosa was incubated with 1 mL of HPLC mobile phase overnight at 25 °C to extract the retained DA, which was then quantified via HPLC analysis.

### 2.8. Cytotoxicity Assessment in Olfactory Ensheathing Cells (OECs) and in SH-SY5Y

OECs, grown as previously described [33], or SH-SY5Y, also grown as previously described [34], were plated in a 96-well plate (BD, USA) at a density of 1 × 10^4^ per well in 200 μL of complete medium and incubated overnight to allow cell attachment. Then, the culture medium was replaced with 200 μL of fresh medium containing different dilutions of GSE-ads-DA-SLNs (final concentrations of dopamine: 75 µM, 50 µM, 25 µM and 12.5 µM, corresponding, respectively, to 0.76 µg/mL, 0.51 µg/mL, 0.25 µg/mL and 0.13 µg/mL of GSE) and DA-co-GSE-SLNs (final concentrations of dopamine: 75 µM, 50 µM, 25 µM and 12.5 µM, corresponding, respectively, to 0.041 µg/mL, 0.027 µg/mL, 0.014 µg/mL and 0.007 µg/mL of GSE), DA (75 µM, 50 µM, 25 µM and 12.5 µM), GSE (0.041 µg/mL, 0.027 µg/mL, 0.014 µg/mL and 0.007 µg/mL) and plain SLN (33 μL, 66 μL, 132 μL and 200 μL) in complete medium (final volume per well was 200 μL), and in these conditions, cells were grown at 37 °C for 24 h. Volumes of SLNs were chosen to obtain the same lipid amounts of the other preparations. Triton-X100-treated cells (0.1%) were used as a positive control. Then, after removing the medium, the cells were incubated with resazurin solution according to the manufacturer’s protocol (Biotium, Fremont, CA, USA [35,36,37]. After incubation, the fluorescence of solubilized resorufin was measured by a FLUOstar Omega microplate reader (BMG Labtech, Ortenberg, Germany) (530 nm excitation; 590 nm emission). Each experiment was performed three times.

### 2.9. Evaluation of Uptake by OECs and SH-SY5Y

OECs or SH-SY5Y were plated on 6-well plates at 2 × 10^5^ cells and, 24 h later, different doses of both GSE-ads-DA-FITC-SLNs and DA-FITC-co-GSE-SLNs were added to the cells. After different times of treatment (2 h, 4 h and 24 h for OECs; 24 h, 48 h and 72 h for SH-SY5Y), cells were washed two times with PBS, harvested by treatment with trypsin/ethylenediaminetetraacetic acid, and, after blocking the trypsin by FBS, cell centrifugation occurred at 350× *g* for 5 min and the obtained pellet was finally resuspended in PBS. Cells were analyzed by a FACSCalibur^TM^ apparatus (Becton-Dickinson, San Jose, CA, USA). Fluorescence for FITC was measured at the wavelength of 525 nm, determining the percentage of FITC-positive cells after setting the gating on 99% and by subtracting the fluorescence of untreated control cells. Ten thousand cells were examined in each experiment, evaluating the percentage of positive cells for FL1H fluorescence and the relative mean fluorescence per cell (MFI). The experiments were performed in triplicate.

### 2.10. Statistics

Statistical analyses were carried out with Prism v. 5.0 (GraphPad Software Inc., La Jolla, CA, USA). Data were expressed as mean ± SD. Multiple comparisons were based on one-way analysis of variance (ANOVA), with the either Bonferroni’s or Tukey’s post hoc tests, and differences were considered significant when *p* < 0.05. For cell viability and internalization experiment’s statistical significance was evaluated with a two-tailed unpaired Student’s *t*-test. Significant differences were obtained when *p* < 0.05.

## 3. Results

### 3.1. Preparation and Characterization of SLNs

Table 1 shows the main physicochemical properties of the DA/GSE SLNs prepared for this study, following the melt homogenization method [19]. In comparison to plain SLNs taken as control, they were found to be significantly bigger in size (Table 1). In details, DA-co-GSE-SLNs were the smallest ones (187 ± 4 nm), whereas, due to the adsorption of GSE or due to the loading of fluorescent molecules such as FITC or DA-FITC, higher mean diameters were obtained for the other SLNs, ranging from 237 to 297 nm. All investigated SLNs should possess a broad and/or multimodal size distribution, as suggested by their high PDI values ranging from 0.46 to 0.62.

Besides the average hydrodynamic diameters obtained by PCS shown in Table 1, the TEM morphological analysis of the unlabeled SLNs at the dried state provided the average diameter by this optical method, which, as expected, resulted into lower than that by PCS (Figure 1 and Appendix A). In particular, low-contrast spheroidal particles were detected, as expected for organic nanomaterials. The estimation of their size was carried out using a manual sizing method, involving the characterization and the statistical analysis on the size distribution of as many particles as possible (*n* > 200). It was confirmed by TEM microscopy that the particle size falls in the same range outlined by the PCS data of Table 1 together with broad size distribution.

Zeta potential measurements all provided slightly negative values due to the negatively charged lipid Gelucire^®®^ 50/13. As reported in Table 1, concerning zeta potential values, all SLNs were statistically different from plain SLNs, exhibiting external surface charges ranging between −1.0 and −7.8 mV. In terms of DA/GSE content, in the case of unlabeled SLNs, namely, DA-co-GSE-SLNs and GSE-ads-DA-SLNs, the E.E. % of DA was quite similar (i.e., close to 60%), whereas E.E. of GSE% in DA-co-GSE- SLNs and GSE-ads-DA-SLNs was 10% and 57%, respectively. Concerning fluorescent SLNs, statistically significant differences were seen in terms of mean diameter only in the case of DA-co-GSE-SLNs vs. DA-FITC-co-GSE-SLNs (*p* ≤ 0.001). Furthermore, the highest percentages of E.E. % in FITC were found when the dye FITC was covalently bound to DA rather than in the case of the already studied FITC-SLNs [21], where the fluorescent marker FITC was physically entrapped in the lipid Gelucire^®®^ 50/13. As detailed in Table 1, for GSE-ads-DA-FITC SLNs, the E.E. % in DA was lower than the corresponding unlabeled SLNs, namely, GSE-ads-DA-SLNs (35 ± 1% vs. 65 ± 6%, respectively).

### 3.2. Physical Stability

Incubation of DA-co-GSE-SLNs and GSE-ads-DA-SLNs suspensions at 4 °C, 25 °C and 37 °C for different time intervals allowed us to determine the physical stability of the nanocarriers, and the results are reported in Figure 2. At 37 °C, for both types of particles, mean diameters increased over time, and, precisely, particle size doubling was detected at the latest time points (Figure 2a, * *p* ≤ 0.01). For particles incubated at 25 °C, DA-co-GSE-SLNs leading to black precipitated only after one week, but not at previous time points. However, for GSE-ads-DA-SLNs, starting from three days of incubation, aggregates were detected, although no visual changes were seen (Figure 2b). Furthermore, at 4 °C, GSE-ads-DA-SLNs showed grey aggregate formation after 2 months of storage at this temperature (Figure 2c), and, moreover, statistically significant differences in particle size appeared at 1, 2, 4 and 8 weeks (*p* ≤ 0.001). On the other hand, for DA-co-GSE-SLNs, size enlargement detected at 4 °C was statistically relevant at 8 and 12 weeks of incubation (*p* ≤ 0.001). We did not observe either aggregation or color modification under incubation at 4 °C of DA-co-GSE-SLNs. It should be noted that, in this paper, besides particle size, we did not evaluate other stability-influencing parameters such as surface charge and drug content in SLNs so as to draw reliable conclusions about the stability profile of these nanocarriers. This is an issue which can be the subject of a future investigation.

### 3.3. Solid State Studies

#### 3.3.1. FT-IR Spectroscopy

FT-IR spectra of the described freeze-dried SLNs are reported in Figure 3A. Notably, both DA-co-GSE-SLNs and GSE-ads-DA-SLNs did not exhibit any typical bands of pure DA and pure GSE between 1615 and 1584 cm^−1^ and at 1609 cm^−1^, respectively (Figure 3A curves (a) and (b)). On the other hand, either DA-co-GSE-SLNs and GSE-ads-DA-SLNs showed the stretching vibration of carbonyl group of Gelucire^®®^ 50/13 at 1738 cm^−1^ (Figure 3A curves (d) and (e)) together with the stretching vibration of -OH groups at the wavenumbers higher than 3477 cm^−1^. It is well known that the absorption bands at 3431 cm^−1^ and 1734 cm^−1^ belong to partially hydrated Gelucire^®®^ 50/13, and in the FT-IR spectrum of plain SLNs, the former band was shifted at 3477 cm^−1^ (Figure 3A curve (c)), [19,38]). For both DA-co-GSE-SLNs and GSE-ads-DA-SLNs, a less crystalline state of DA was detected, although Gelucire^®®^ 50/13 bands are predominant in the spectra (Figure 3A curves (d) and (e)).

#### 3.3.2. DSC Analysis

DSC thermograms of both DA-co-GSE-SLNs and GSE-ads-DA-SLNs suggested that none of the endothermic peaks attributable to the melting of pure DA and pure GSE (at 250 °C and 160 °C, respectively) were revealed (Figure 3B curves (c)–(e)). On the other hand, the amorphous state of plain SLNs was detected for DA-co-GSE-SLNs and GSE-ads-DA-SLNs. As shown in Figure 3B curve (b), a small peak at about 105 °C was observed, and it was attributable to the shifted GSE melting point, probably suggesting an external location of the antioxidant mixture.

### 3.4. Franz Diffusion Cell Permeation

In Figure 4 (panels A and B), the permeation profiles via Franz diffusion cells of DA-co-GSE-SLNs and GSE-ads-DA-SLNs vs. unencapsulated DA are shown. When free DA permeated, almost 5 mg was released in the SNF-receiving medium of the Franz cell in 24 h (Figure 4A). On the other hand, the neurotransmitter DA, permeated with a sustained profile either from DA-co-GSE-SLNs and GSE-ads-DA-SLNs, reached into the receiving compartment the highest levels equal to to 3.1 μg and 1.7 μg in 24 h, respectively (Figure 4B). The apparent permeability coefficients (*P*_app_) were also calculated referring to DA, unencapsulated and from SLNs permeating across the nasal porcine mucosa and are shown in Table 2. Notably, *P*_app_ of unencapsulated DA was equal to 9.1 × 10^−3^ ± 1 × 10^−4^ cm/s, leading to a marked reduction when it was calculated for DA-co-GSE-SLNs (i.e., 4.7 × 10^−7^ ± 6.0 × 10^−8^ cm/s) and for GSE-ads-DA-SLNs (i.e., 3.8 × 10^−7^ ± 2.4 × 10^−8^ cm/s). Furthermore, no change in color of the receiving medium was observed, both at the early and late incubation time points, irrespective of the tested SLN formulation, suggesting that no autoxidation of DA occurred throughout the experiments. After the extraction of DA from nasal porcine mucosa, 0.03 μg/cm^2^ and 4.4 μg/cm^2^ of oxidized DA for DA-co-GSE SLNs and GSE-ads-DA-SLNs were found, respectively. In the HPLC chromatograms acquired after extraction, no peak ascribable to intact DA was detected, but the HPLC peak appearing was attributed to the oxidized neurotransmitter. The attribution of the HPLC peak to the oxidized DA was also in good agreement with the black areas visualized on the membrane, irrespective of the SLN considered (Appendix A), when the porcine nasal mucosa was removed from the Franz cell at the end of the experiment. On the other hand, control porcine nasal membranes (i.e., in the absence of any incubated sample) did not show any black area at the end of the experiment (Appendix A).

### 3.5. Cytobiocompatibility of DA/GSE SLNs in OECs

To assess the biocompatibility of DA/GSE SLNs with a cell type relevant for the NTB delivery, the effect of different doses of GSE-ads-DA-SLNs and DA-co-GSE-SLNs (respectively, 12.5, 25, 50 and 75 µM DA) on the viability of OECs was studied after 24 h of treatment. Results shown in Figure 5 indicate that all the doses of both formulations are well-tolerated.

### 3.6. Uptake of SLNs by OECs

The residence time in the nasal cavity is usually of few hours, thus, the uptake of SLNs by OECs was assessed after 2 h, 4 h and 24 h of treatment. The percentage of positive OECs and the relative mean fluorescence per cell were analyzed by flow cytometry by incubating cells with different doses of both GSE-ads-DA-FITC-SLNs (Figure 6a,b) and DA-FITC-co-GSE-SLNs (Figure 6c,d) used in the previous experiments of cytotoxicity (respectively, 12.5, 25, 50 and 75 µM DA). Results show that after 2 h of treatment with both formulations, there was a small percentage of FITC-positive cells that slightly increased after 4 h. After 24 h, the percentage of positive cells was higher after the treatment with DA-FITC-co-GSE-SLNs in comparison with GSE-ads-DA-FITC-SLNs. These data indicate that the longer the residence time of SLNs in the nasal cavity, the higher the uptake by OECs, and that the internalization process is better for DA-co-GSE-SLNs.

### 3.7. Cytobiocompatibility of DA/GSE SLNs in SH-SY5Y Cells

The SH-SY5Y cell viability was studied 24 h after incubation with different doses of both GSE-ads-DA-SLNs and DA-co-GSE-SLNs (respectively, 12.5, 25, 50 and 75 µM DA). Results in Figure 7 indicate that 12.5 µM and 25 µM doses are well-tolerated for both formulations, while a significant reduction in viability was observed at 50 µM for GSE-ads-DA-FITC-SLNs. At these concentrations, the DA-co-GSE-SLNs were still safe, while they determined a small but significant reduction in viability at 75 µM.

### 3.8. Uptake of SLNs by SH-SY5Y Cells

To verify that SH-SY5Y cells could uptake SLNs employed in this study for prolonged times, the percentage of positive SH-SY5Y cells and the relative mean fluorescence per cell were analyzed by flow cytometry after 24 h, 48 h and 72 h of treatment, with different doses of both GSE-ads-DA-FITC-SLNs (Figure 8a,c) and DA-FITC-co-GSE-SLNs (Figure 8b,d) used in the previous experiments of cytotoxicity (respectively, 12.5, 25, 50 and 75 µM DA). Results show that the percentage of cells that internalize the SLNs enhanced by increasing the time of treatment reaching higher values for DA-FITC-co-GSE-SLNs in comparison with ads-DA-FITC-SLNs; similarly, mean fluorescence intensity enhanced as the contact time with the nanoparticles increased.

## 4. Discussion

In recent years, for Alzheimer disease (AD) and PD, the multidrug therapy approach has attracted attention in order to administer synergic drug substances capable of limiting the progression of these pathologies. This is thanks to the fact that their mechanisms of action, although different, can be considered complementary to each other, and, thus, the combination results are more beneficial than each drug substance being used alone. In the literature, some combinations of active principles delivered by conventional or modified drug delivery systems were reported to be tested for anti-PD (or anti-AD) therapies.

Thus, resveratrol associated with grape-extract-loaded SLNs evidenced synergic performances in AD [39]. Similarly, the combination of quercetine/piperine has recently been evaluated in parkinsonian rats, showing neuroprotective effect [40]. Such hints from the literature motivated our interest to combine DA and GSE in SLN carriers to be intranasally administered with the aim to restore neuron functionality in the brain of parkinsonian patients. For the purpose of encapsulating two hydrophilic active principles in SLNs, we used the melt homogenization method [22]. Concerning this preparative method, we engaged in evaluating the scope and limitations to gain insights into its practical application [12]. Thus, we showed that the method works in a satisfactory way when a single drug is to be encapsulated and also with hydrophilic drugs exhibiting large differences in chemical structure and molecular weight [12,18,21]. Hence, the melt homogenization method could represent a simple alternative to the double-emulsion method for the preparation of hydrophilic, drug-loaded SLNs with satisfactory encapsulation efficiency and with the advantage of being organic solvent free.

We started to prepare such nanocarriers previously by the mentioned method [19], and, as a continuation, in the present work, we have paid our attention on two types of formulations. The first, herein described for the first time, led to SLNs where, during preparation, DA and GSE were both poured in acetic acid phase (i.e, DA-co-GSE-SLNs), whereas in the other approach previously reported [19], once DA-SLNs were at first achieved, then GSE was physically adsorbed on them (i.e., GSE-ads-DA-SLNs). Actually, preliminary studies aiming at increasing the adsorbed GSE *cargo* onto preformed SLNs have also evidenced that, at higher concentrations than 1 mg/mL, GSE formed suspensions rather than clear solutions. Additionally, when DA was introduced in the aqueous phase at concentrations higher than 20 mg/mL, then grey SLNs were recovered, maybe due to the oxidation of the neurotransmitter present, so hindering the use of such concentrations. Based on these experiments, to achieve the formulation of DA-co-GSE-SLNs rather than GSE-ads-DA-SLNs, the predispersion of GSE and homogenization were required (see Section 2.2.1) in order to obtain small drops, which could be suitable to form the emulsion with the lipid Gelucire^®®^ 50/13. Moreover, particles produced with the two approaches were not only significantly different in terms of mean diameter (*p* ≤ 0.001) and in terms of zeta potential (Table 1), since both formulations of GSE-ads-DA-SLNs and DA-co-GSE-SLNs provided external surface charges ranging between −7.8 mV and −4.1 mV, respectively, and both were statistically different from the negative zeta potential of plain SLNs. In addition, the data of Table 1 clearly show a limitation of the preparative method used by us when it is employed to coencapsulate two hydrophilic active principles or when one of them is present in high amounts (such as the neurotransmitter DA at 20 mg as starting material) in the self-emulsifying matrix (Gelucire^®®^ 50/13). Under these forced conditions, a consistent increase in the PDI value of the resulting particles is clearly observed, ranging from 0.49 to 0.59, suggesting a very broad size distribution. Indeed, the simultaneous loading of two hydrophilic substances, as well as the encapsulation of single hydrophilic compounds at high doses in the lipid matrix of SLNs, is not a simple task. The overcoming of the observed limitations represents a challenge for scientists in the field and, incidentally, we are involved in refining the original melt homogenization method, which has so far followed and the results of these studies ì will be reported in due course.

The observed results reported in Table 1 fit with our general model of Gelucire^®®^ 50/13-SLNs, consisting of a hydrophilic shell of polyoxyethylene chains of Gelucire^®®^ 50/13 and the cosurfactant (Tween^®®^ 85), where DA and GSE are supposed to be on the external side of the particles or entrapped in the hydrophilic shell, as well as forming a nanoemulsion in the lipid core [21].

Another interesting insight that can be deduced from Table 1 concerns the ratio between E.E.% referring to DA and E.E.% referring to GSE (E.E.% DA/E.E.%GSE), representing an interesting outcome in terms of future perspectives in PD applications of these nanocarriers. Thus, while this ratio was about 1:1 for GSE-ads-DA-SLNs, it changed to about 6:1 for DA-co-GSE-SLNs. Moreover, it is noteworthy that, in the preparation of GSE-ads-DA-SLNs, the E.E.% DA/E.E.% GSE ratio is reversed starting from 20 mg of neurotransmitter instead of from the 10 mg previously employed [19]. Of course, when the amount of the antioxidant agent is equal or prevalent in comparison with that of the neurotransmitter, the former may protect DA on the surface of the nanocarrier from the spontaneous autoxidation reaction in the presence of molecular oxygen, leading to grey-black polymer compounds (e.g., neuromelanins).

Such considerations may help to rationalize the result observed in the course of physical stability studies at 25 °C, where DA-co-GSE-SLNs led to black precipitate formation, while for GSE-ads-DA-SLNs, no visual changes were seen. It is possible that the positive outcome observed with the latter SLNs is related to the antioxidant amount, comparable with that of the neurotransmitter. On the other hand, the result, consisting of a greater physical stability of DA-co-GSE-SLNs from the particle size point of view, may be due, at least in part, to the more negative zeta potential value of these nanocarriers compared with GSE-ads-DA-SLNs (i.e., −7.8 ± 0.4 mV vs. −4.1 ± 0.1 mV, respectively).

Concerning the physicochemical characterization of SLNs, freeze-dried samples of DA-co-GSE-SLNs and GSE-ads-DA-SLNs were examined by FT-IR spectroscopy, and the thermal analysis is reported in Figure 3. Altogether, the FT-IR spectral patterns of the two SLNs, as well as the DSC thermograms, are indicative of a marked reduction in the crystalline state of both DA and GSE compounds. Indeed, also from the DSC thermograms reported in Figure 3B,C, no endothermic peak ascribable to pure DA and/or GSE was revealed, thus reinforcing the conclusion about the reduction in the crystalline state.

In view of the application to NTB delivery, in this work, two types of fluorescent SLNs were produced, namely, FITC-SLNs and DA-FITC SLNs, in the presence or in the absence of GSE. Indeed, FITC-SLNs physically entrapped the dye FITC, as previously studied by us [21], whereas DA-FITC was synthetized according to Carta et al.’s protocol [20] in order to connect via the formation of thioamide bonding the neurotransmitter to FITC dye. For particle tracking in OECs, FITC-SLNs were selected because, once SLNs have permeated the nasal mucosa, the fluorescent signals detected in OECs could undoubtedly demonstrate that the olfactory pathway was followed. Additionally, internalization experiments with FITC-SLNs, rather than DA-FITC SLNs in the presence or in the absence of GSE, could be usefully compared with future uptake studies, where, for instance, semisolid dosage forms incorporating FITC-SLNs are exposed to OECs in order to evidence the formulation taken up in the highest amount which, hence, could merit in vivo testing for patient applications. From a physicochemical viewpoint, GSE-ads-DA-FITC-SLNs were endowed with a higher negative surface charge in comparison to unlabeled GSE-ads-DA-SLNs, and the content of the neurotransmitter DA was found to be almost a half of the corresponding nonfluorescent GSE-ads-DA-SLNs (Table 1). We believe that replacing DA with FITC-DA could slightly edit the internal/external structure of the SLNs, but further investigations focusing on the composition of the particles are required to clarify this aspect. For a better understanding of the nasal absorption mechanisms involving SLNs, among ex vivo permeation approaches [27,41], Franz diffusion cells were selected and mounted with porcine nasal mucosa for 24 h (Figure 4), being a convenient tool to explore the ex vivo transport because no fluorescent/radiolabeled compound nor specific pharmacodynamic in vivo tests in animal models were required [42]. As reported in Table 2, unencapsulated DA showed a high *P*_app_ (i.e., 9.1 × 10^−3^ ± 1 × 10^−4^ cm/s), similar to some water-soluble drugs with low molecular weight that have been already found to have a high flux permeation value in a model of excised hairless rat skin [43]. In such a model, indeed, 458 μg/cm^2^/h was the DA flux calculated by the authors [43]. In the present work, when *P*_app_ of DA was calculated after permeation of the neurotransmitter from SLNs, a significant reduction in the corresponding free form was evidenced, as a consequence of the loading into the nanosystems. Indeed, high amounts of permeated DA could not be expected because the lipophilic carrier of SLN can exert an enhancement role for hydrophobic drugs crossing the nasal mucosa. In the literature, in the presence of goat nasal mucosa for a lipophilic drug such as ferulic acid, the remarkable enhancement of permeation due to SLNs was ascribed to their lipidic architectural constitution [44]. Another example is the enhancement of the crossing of the porcine nasal mucosa of a hydrophobic drug substance such as opiorphin by liposomes [27], similarly to the enhancement effect due to the so-called “elastic liposomes” already assessed for other lipophilic agents [45].

In details, due to the fact that our SLNs administer two agents rather than only one, for each substance the permeation process should be influenced (i) by the nanocarrier SLN and (ii) by the other active compound. In fact, the reduction in DA *P*_app_ value leads to distinguish between the amount permeated in 24 h from DA-co-GSE-SLNs vs. GSE-ads-DA-SLNs (3.1 μg vs. 1.7 μg, respectively, Figure 4b). Currently, we could assess that some DA was retained by the porcine nasal mucosa. In fact, after extraction from the porcine nasal mucosa as described in Section 2.7, black areas onto the mucosa were visualized (Appendix A), inducing us to consider that, after retention of DA in the nasal mucosa metabolization of the neurotransmitter, degradation occurred due to the enzymes of the mucosa. Such hypothesis was also corroborated by the HPLC chromatograms where the peak attributable to oxidized DA was shown. Indeed, via the apparatus of Franz cell coupled to HPLC analysis we mainly aimed at calculating the amounts permeated of DA from the donor to the acceptor compartment in order to ascertain that SLNs quantitatively deliver DA and, consequently, NTB delivery can be reasonably predicted. Overall, the presence of GSE represents a benefit for DA permeation because, the prevention of DA autoxidation exerted by this mixture was achieved for both types of SLNs as no color change was seen in the withdrawals during the experiments with Franz diffusion cells, irrespective of the SLNs tested. Therefore, we can consider promising both SLNs in terms of ex vivo nasal permeation performances because, although DA is delivered at the maximum amount of 1.7 μg when GSE-ads-DA-SLNs are incubated, from the pharmacological viewpoint, the intact neurotransmitter is considered a powerful substance which can elicit a biological effect also at low doses released.

On the other hand, when SLNs containing DA/GSE were incubated with neuronal SH-SY5Y, as shown in Figure 7, GSE-ads-DA-SLNs were found to be more toxic than DA-co-GSE-SLNs. Our hypothesis is that the high levels of GSE associated to this type of SLN could induce a high depletion of ROS species in the cells, so lowering the same defense system represented by the ROS production. Recently, for several body districts it has been assessed that ROS compounds should be present in some extent for the proper cell or tissue function [46,47,48] and a delicate balance between antioxidant supplementation and ROS presence is crucial for living cells [49]. Furthermore, flow cytometry was employed for elucidating the uptake in the neuronal SH-SY5Y cells and higher percentages of internalization for DA-FITC-co-GSE-SLNs rather than for GSE-ads-DA-FITC-SLNs were assessed (Figure 8). Being particle sizes of GSE-ads-DA-FITC-SLNs and DA-FITC-co-GSE-SLNs statistically different (Table 1), it could not be ruled out that the higher capability of internalization towards DA-FITC-co-GSE-SLNs is related to the dimensions of the nanocarriers rather than to the morphology, which seems quite similar between the two types of SLNs as derived from TEM visualization (Figure 1 and Appendix A).

## 5. Conclusions

The need to supply the missing neurotransmitter DA together with antioxidant agents to the brain of PD patients led us to develop a nanoparticulate delivery system, where in vitro characterization was performed in terms of particle size, zeta potential, morphological visualization and physical stability. Furthermore, the permeation through the porcine nasal mucosa allowed us to assess that DA from both types of SLNs were delivered in the nasal compartment, whereas the higher capability of neuronal SH-SY5Y to take up DA-FITC-co-GSE-SLNs than GSE-ads-DA-FITC-SLNs was also shown, thanks to flow cytometry studies. Overall, the internalization of SLNs is encouraging for proceeding in the investigations, with the idea in mind to address them, via the use of semisolid dosage forms, in order to achieve PD treatment through a NTB approach.

## Figures and Tables

**Figure 1 pharmaceutics-15-00881-f001:**
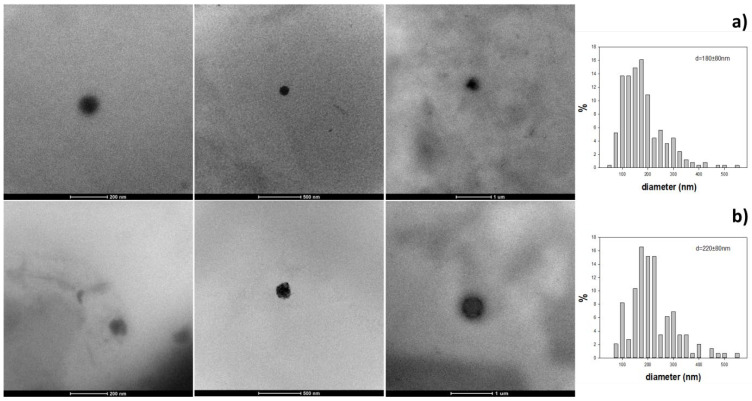
Typical transmission electron microscopy images of individual SLNs in the dried state and under different magnifications (from left to right, the scale bar corresponds to 200, 500 and 1000 nm, respectively). Upper row (**a**): DA-co-GSE-SLNs, average size 180 ± 80 nm; lower row (**b**): GSE-ads-DA-SLNs, average size 220 ± 80 nm; in both cases: *n* > 200. Plot (**a**): Size distribution from TEM observations of DA-co-GSE-SLNs; Plot (**b**): Size distribution from TEM observations of GSE-ads-DA-SLNs. For further details, see Appendix A.

**Figure 2 pharmaceutics-15-00881-f002:**
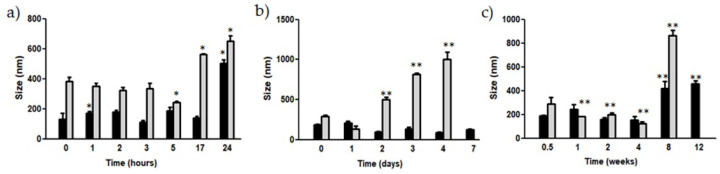
Particle size variation of SLNs after incubation at (**a**) 37 °C for 24 h; (**b**) 25 °C for one week; (**c**) 4 °C for three months. DA-co-GSE-SLNs (black bars). GSE-ads-DA-SLNs (grey bars). * *p* ≤ 0.01; ** *p* ≤ 0.001. Controls were mean particle sizes of DA-co-GSE-SLNs and GSE-ads-DA-SLNs, as reported in Table 1.

**Figure 3 pharmaceutics-15-00881-f003:**
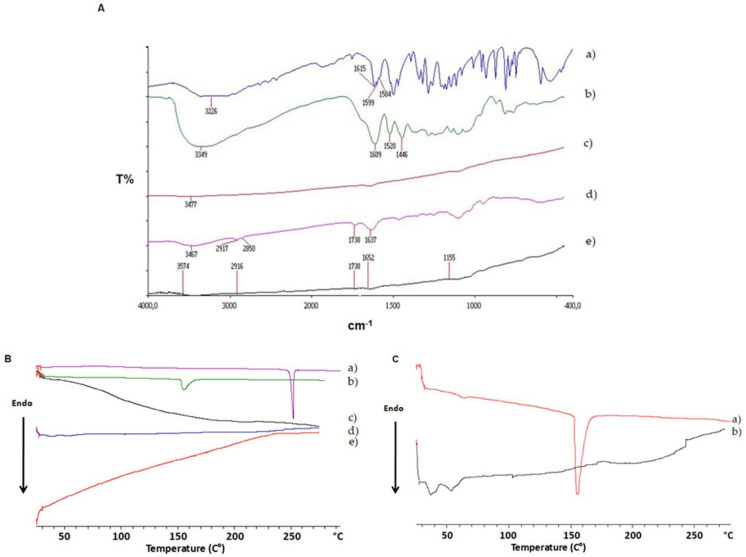
Panel (**A**): FT-IR spectra of pure DA (**a**), pure GSE (**b**), plain SLNs (**c**), DA-co-GSE-SLNs (**d**) and GSE-ads-DA-SLNs I (**e**). Panel (**B**): DSC profiles of pure DA (**a**), pure GSE (**b**), plain SLNs (**c**), DA-co-GSE-SLNs (**d**) and GSE-ads-DA-SLI (**e**); Panel (**C**): Details of DSC thermal curves of pure GSE (**a**) and GSE-ads-DA-SLNs (**b**).

**Figure 4 pharmaceutics-15-00881-f004:**
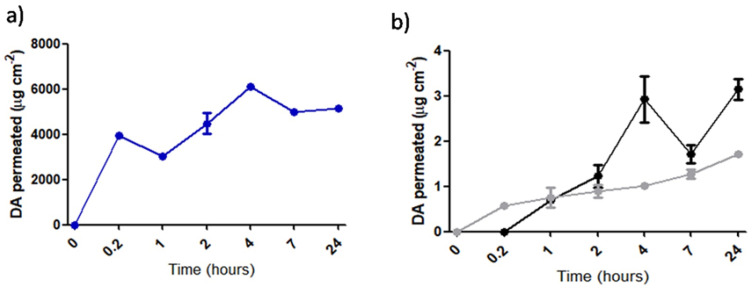
Franz diffusion cell permeation profiles of (**a**) unencapsulated DA; (**b**) DA from DA-co-GSE-SLNs and from GSE-ads-DA-SLNs; blue: Unencapsulated DA; black: DA-co-GSE-SLNs; grey; GSE-ads-DA-SLNs. Results are means ± SD (*n* = 5). For further details, see Appendix A.

**Figure 5 pharmaceutics-15-00881-f005:**
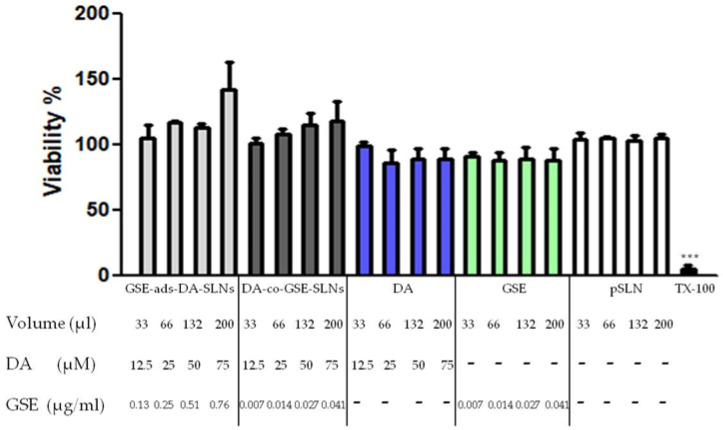
Cytobiocompatibility of GSE-ads-DA-SLNs (light grey bars), DA-co-GSE-SLNs (dark grey bars), DA (blue bars), GSE (light green bars) and plain SLN (pSLN, white bars) on OECs. OECs were challenged with the different formulations or free substances at the indicated volumes and concentrations for 24 h. pSLN were used at equivalent volumes. After 24 h of incubation, resazurin test was performed to assess cell viability. TX-100 (0.1% Triton X-100) denotes positive control. Data are expressed as the mean ± SD of two experiments conducted each in six replicates. Untreated cells (not shown) provided 100% of viability. *** *p* < 0.001 for TX-100 treated cells vs. untreated cells.

**Figure 6 pharmaceutics-15-00881-f006:**
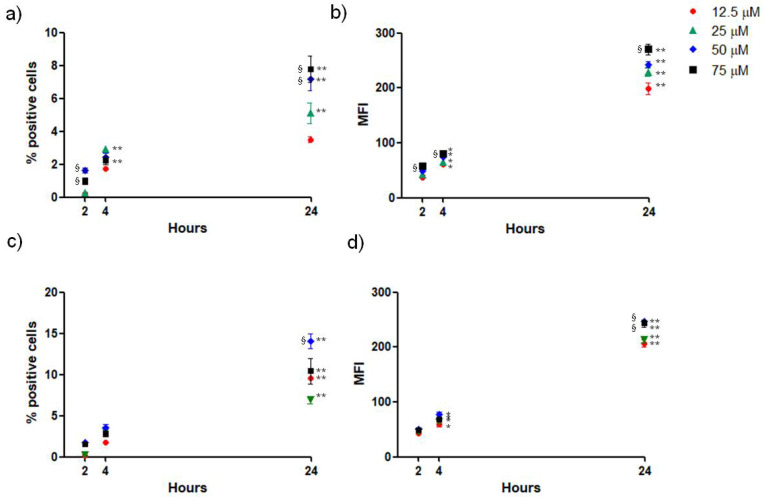
Uptake of SLN formulations by OECs. Cells were incubated with different doses of GSE-ads-DA-FITC-SLNs (**a**,**b**) and DA-FITC-co-GSE-SLNs (**c**,**d**) for 2 h, 4 h and 24 h and then evaluated for percentage of positive fluorescent cells (**a**,**c**) and relative mean fluorescence (MFI) (**b**,**d**) by flow cytometry. Data are expressed as the mean ± SD of three experiments, each conducted in duplicate. * *p* < 0.05 and ** *p* < 0.01 indicate statistical differences between the same dose at 24 h and 4 h vs. 2 h; ^§^
*p* < 0.05 indicates statistical differences between 75 µM and 50 µM doses vs. 12.5 µM dose at the same time.

**Figure 7 pharmaceutics-15-00881-f007:**
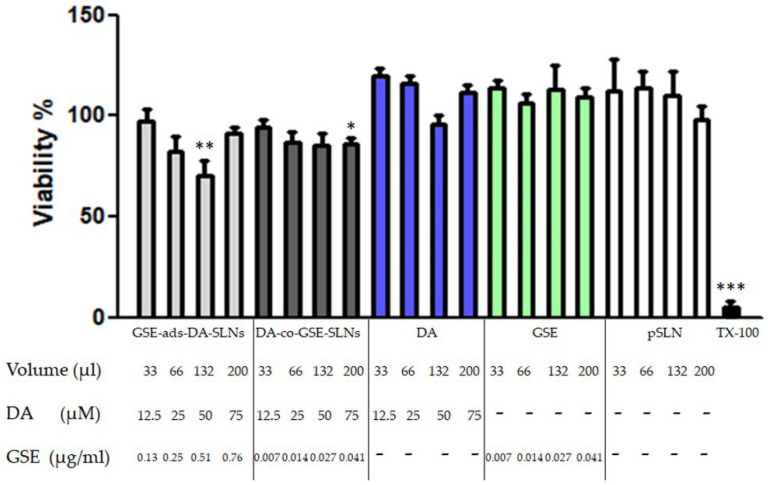
Cytocompatibility of GSE-ads-DA-SLNs (light grey bars), DA-co-GSE-SLNs (dark grey bars), DA (blue bars), GSE (light green bars) and plain SLN (pSLN, white bars). SH-SY5Y cells were challenged with the different formulations or free substances at the indicated volumes and concentrations for 24 h. pSLN was used at equivalent volumes. After 24 h of incubation, a resazurin test was performed to assess cell viability. TX-100 (0.1% Triton X-100) denotes positive control. Data are expressed as the mean ± SD of two experiments, each conducted in six replicates. Untreated cells provided 100% of viability. * *p* < 0.05, ** *p* < 0.01, *** *p* < 0.001.

**Figure 8 pharmaceutics-15-00881-f008:**
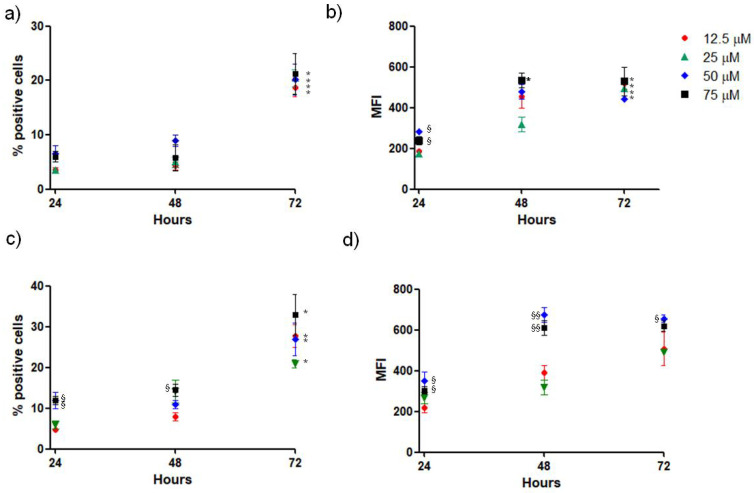
Uptake of SLN formulations by SH-SY5Y cells. Cells were incubated with different doses of GSE-ads-DA-FITC-SLNs (**a**,**b**) and DA-FITC-co-GSE-SLNs (**c**,**d**), considering the DA concentration for 24 h, 48 h and 72 h, and then evaluated for percentage of positive fluorescent cells (**a**,**c**) and relative mean fluorescence (MFI) (**b**,**d**) by flow cytometry. Data are expressed as the mean ± SD of three experiments, each conducted in duplicate. * *p* < 0.05 indicates statistical differences between the same dose at 24 h and 4 h vs. 2 h; ^§^
*p* < 0.05 and ^§§^
*p* < 0.01 indicate statistical differences between 75 µM and 50 µM doses vs. 12.5 µM dose at the same time.

**Table 1 pharmaceutics-15-00881-t001:** Physicochemical properties of SLNs prepared. Mean ± standard deviation of at least eight replicates is reported.

Formulation	Size (nm)	PDI ^a^	Zeta Potential (mV)	E.E. DA (%)	E.E. GSE (%)	E.E. FITC (%)
DA-co-GSE-SLNs	187 ± 4 **	0.49 ± 0.04	−4.1 ± 0.1 **	62 ± 4	10 ± 0	-
GSE-ads-DA-SLNs	287 ± 15 **	0.53 ± 0.01	−7.8 ± 0.4 **	6 5 ± 6	57 ± 8	-
FITC-SLNs ^b^	275 ± 9 **	0.53 ± 0.08	−2.9 ± 0.2 **	-	-	55 ± 4
DA-FITC-SLNs	237 ± 6 **	0.56 ± 0.03	−1.0 ± 0.4 **	88 ± 2	-	99 ± 2
DA-FITC-co-GSE-SLNs	297 ± 25 **	0.59 ± 0.04	−3.6 ± 0.1 **	82 ± 4	23 ± 1	94 ± 6
GSE-ads-DA-FITC-SLNs	266 ± 12 **	0.57± 0.04	−8.1 ± 0.1 **	35 ± 1	42 ± 2	96 ± 1
Plain SLNs ^b^	141 ± 11	0.35 ± 0.17	−9.7 ± 0.8	-	-	-

Plain SLNs were taken as control for statistical evaluation. ^a^ PDI: polydispersity index. ^b^ From reference [21]. ** *p* ≤ 0.001.

**Table 2 pharmaceutics-15-00881-t002:** Apparent permeability coefficient (*P*_app_) of DA, unencapsulated and from SLNs, across excised nasal porcine mucosa. Data are mean ± standard deviation of five replicates (*n* = 5). ** *p* ≤ 0.001 calculated with respect to *P*_app_ of unencapsulated DA.

Formulation	*P*_app_ DA (cm/s)
DA-co-GSE-SLNs	4.7 × 10^−7^ ± 6.0 × 10^−8^ **
GSE-ads-DA-SLNs	3.8 × 10^−7^ ± 2.4 × 10^−8^ **
Unencapsulated DA	9.1 × 10^−3^ ± 1 × 10^−4^

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
