# Peer review of "Combined Dopamine and Grape Seed Extract-Loaded Solid Lipid Nanoparticles: Nasal Mucosa Permeation, and Uptake by Olfactory Ensheathing Cells and Neuronal SH-SY5Y Cells [Author-notes fn1-pharmaceutics-15-00881]"

_pharmaceutics, 2023, doi:10.3390/pharmaceutics15030881_

Round 1

Reviewer 1 Report (Previous Reviewer 2)

After revising the manuscript, I suggest that this paper to be pubilished in Pharmaceutics.

Author Response

We are grateful to the Reviewer for his/her positive decision

Reviewer 2 Report (Previous Reviewer 4)

Authors have revised the paper by omitting studies on sodium alginate gels and focusing on SLNs as carriers for dopamine and grape seed extract to be applied nasally, which may be an acceptable approach.

However, there are still some issues left to be addressed by the authors to improve the quality of the manuscript.

2.6. Assessment of physical stability of SLNs – apart from particle size, no other stability parameters were evaluated, like zeta-potential or drug content in the formulation, although they are crucial to derive the conclusion on SLN stability profile.

Ex vivo SLN permeation studies; Page 6, lines 250-253; „For the experiments, the anterior surface of the mucosa was faced towards the donor compartment (effective diffusion area = 0.6 cm2), and freeze-dried DA-co-GSE-SLNs (or GSE-ads-DA-SLNs) were resuspended in 100 uL of SNF (pH 5-6) (29, 30) and placed in the donor compartment.“ In the previous correspondence within the revision process, authors explained using the freeze-dried product in this study by describing advantages of nasal powders. If authors are developing nasal powders then dried formulations need to be characterized/optimised in terms of rheological properties (powder flow properties) to confirm they can be adequately applied nasally. If freeze-dried powders are aimed to be resuspended prior to administration, knowing particle size and zeta potential upon resuspension and before permeation studies would be advantageous enabling the discussion of obtained results in relation to relevant SLN size and zeta-potential.

Did the authors check the recovery by adding together the DA amount found in the receptor medium, within the mucosa, and the fraction retained in the donor compartment in the end of experiment?

Additional revision of English is needed (e.g.lines 29-30; "Mean diameter of DA...resulted in..."; 102-106; ” The relevance of GSE as effective antioxidant agent was recently showed by some recent papers in which it was evidenced the enhanced antioxidant and antibacterial properties of such natural mixture…”).

Author Response

  1. Authors have revised the paper by omitting studies on sodium alginate gels and focusing on SLNs as carriers for dopamine and grape seed extract to be applied nasally, which may be an acceptable approach.

However, there are still some issues left to be addressed by the authors to improve the quality of the manuscript.

2.6. Assessment of physical stability of SLNs – apart from particle size, no other stability parameters were evaluated, like zeta-potential or drug content in the formulation, although they are crucial to derive the conclusion on SLN stability profile.

  1. We thank the Reviewer for this comment because it allows us to better clarify some our choices made in the course of the study described in this manuscript. It is true that we assessed the physical stability of SLNs only from a particle size point of view but not from other important aspects such as zeta-potential and drug content. Our attention was focused only on the particle size since it is an important parameter that promote the transport of nanoparticles (NPs) across the nasal mucosa , and by sure would play a significant role in intranasal delivery. But this does not mean to decrease the importance of other factors such as the surface charge or the drug content in formulation. It is well known, indeed, that the surface charge of NPs affects their cellular uptake, biodistribution and fate in biological systems. Negatively charged NPs present a faster diffusion in tissues and a higher accumulation in tumour tissues when compared to positively charged NPs [Jo, D.H., et al., Size, surface charge, and shape determine therapeutic effects of nanoparticles on brain and retinal diseases. Nanomedicine, 2015. 11(7): p. 1603-11. 798]. However, we considered the assessment of the physical stability from a surface charge and drug content in formulation an aspect which can be investigated in a next paper. The complete SLN stability profile was out of the primary objectives of the manuscript. We revised the manuscript and, at the end of the paragraph 3.2., we added “It should be noted that, in this paper, besides particle size, we did not evaluate other stability influencing parameters such as surface charge and drug content in SLNs to drawn reliable conclusion about the stability profile of these nanocarriers. This is an issue which can be subject of a future investigation.”
  2. Ex vivo SLN permeation studies; Page 6, lines 250-253; „For the experiments, the anterior surface of the mucosa was faced towards the donor compartment (effective diffusion area = 0.6 cm2), and freeze-dried DA-co-GSE-SLNs (or GSE-ads-DA-SLNs) were resuspended in 100 uL of SNF (pH 5-6) (29, 30) and placed in the donor compartment.“ In the previous correspondence within the revision process, authors explained using the freeze-dried product in this study by describing advantages of nasal powders. If authors are developing nasal powders then dried formulations need to be characterized/optimised in terms of rheological properties (powder flow properties) to confirm they can be adequately applied nasally. If freeze-dried powders are aimed to be resuspended prior to administration, knowing particle size and zeta potential upon resuspension and before permeation studies would be advantageous enabling the discussion of obtained results in relation to relevant SLN size and zeta-potential.
  3. We thank the Reviewer for this comment since it offers the possibility to better clarify some aspects that should be taken into account. We are aimed not at developing nasal powders but our final product is represented by hydrogel incorporating SLNs to minimize mucociliary clearance and to avoid drug loss following intranasal administration. Besides this, it should be also considered that DA is a molecule very sensitive to autoxidation reaction and, therefore, it can be degraded in the presence of oxygen. It forced us to avoid time consuming procedures during manufacturing, handling, and administration even in acid conditions (pH 5-6) where the autoxidation reaction of neurotransmitter is slowed down and hence sample manipulation is possible to same extent. Our choice to develop a hydrogel is based also on the possibility that such formulation further may slow down the autoxidation reaction of the neurotransmitter limiting the direct contact of DA with oxygen. For these reasons, in our study we did not check the mean diameters of nanocarriers after freeze-drying process and therefore we cannot perform the analysis suggested by the Reviewer. However, this is a further issue which can be subject of a future investigation.
  4. Did the authors check the recovery by adding together the DA amount found in the receptor medium, within the mucosa, and the fraction retained in the donor compartment in the end of experiment?
  5. At the end of the permeation experiments, mass balance studies were not carried out to measure the amounts of DA on the surface and inside the membrane. Once again, this choice was obliged for the degradation reaction of the neurotransmitter. Indeed, as mentioned at the end of the paragraph 3.4 of the manuscript “… black areas visualized on the membrane, irrespectively of the SLN considered (Figure S3 a and S3 b), when the porcine nasal mucosa was removed from the Franz cell at the end of the experiment” clearly suggesting that degradation of the neurotransmitter occurred due to the enzymes of mucosa. Under such conditions, some reservations about the significance of recovery tests, in our opinion, can rightly be raised. Indeed, the calculation of the neurotransmitter recovery could not be done properly since the fractions in the receptor medium and in the donor compartment cannot be exactly determined. It induced us to don’t check the recovery at the end of experiments. Being aware of this limitation, in Discussion section we added at Lines 642-645 “Indeed, via the apparatus of Franz cell coupled to HPLC analysis we mainly aimed at calculating the amounts permeated of DA from the donor to the acceptor compartment in order to ascertain that SLNs quantitatively deliver DA and, consequently, NTB delivery can be reasonably predicted”.
  6. Additional revision of English is needed (e.g.lines 29-30; "Mean diameter of DA...resulted in..."; 102-106; ” The relevance of GSE as effective antioxidant agent was recently showed by some recent papers in which it was evidenced the enhanced antioxidant and antibacterial properties of such natural mixture…”).
  7. We apologize for the mistakes occurred and in the revised version, the sentences were corrected as follows. At Lines 29-30 we have corrected: “Mean diameter of DA co-encapsulating GSE SLNs was 187±4 nm”. At Lines 102-106 we wrote: “The relevance of GSE as effective antioxidant mixture was showed in some recent papers, where the enhanced antioxidant and antibacterial properties of such natural mixture with possible application in food packaging [17] as well the oxidative stress decrease in respiratory syndromes [18] were described”

Reviewer 3 Report (New Reviewer)

Trapani et al. explored two approaches to DA/GSE loading in this manuscript: coadministration of DA and GSE in the aqueous phase and physical adsorption of GSE onto premade DA-containing SLNs. Inconsistencies and typos in the language aren't the only problems with the research.
1. For starters, DA-co-GSE-SLNs have a greater PDI than other SLN types (0.49). A typical intranasal nanocarrier has a PDI of less than 0.3. Compared to other intranasal lipid carriers, DA's EE of 62% is rather low. In general, intranasal lipid-based nanocarriers need an EE of 80% or higher (10.1016/j.ejps.2012.05.010). A better option for transporting molecules would be nanostructured lipid carriers (NLCs), making the decision to use SLNs in this instance questionable.

2. Dopamine, a catecholamine neurotransmitter, oxidizes quickly in water. During production, no anti-oxidants were included.

3. In the study, the authors did not use translational mucosal models, which are needed for in vitro testing of drug formulations for intranasal administration.

4. No in vitro release tests were conducted on the finished product.

5. Not enough tests were conducted to support the authors' claims. Large scale testing and characterization are still needed.

Author Response

Trapani et al. explored two approaches to DA/GSE loading in this manuscript: coadministration of DA and GSE in the aqueous phase and physical adsorption of GSE onto premade DA-containing SLNs. Inconsistencies and typos in the language aren't the only problems with the research.

  1. For starters, DA-co-GSE-SLNs have a greater PDI than other SLN types (0.49). A typical intranasal nanocarrier has a PDI of less than 0.3. Compared to other intranasal lipid carriers, DA's EE of 62% is rather low. In general, intranasal lipid-based nanocarriers need an EE of 80% or higher (10.1016/j.ejps.2012.05.010). A better option for transporting molecules would be nanostructured lipid carriers (NLCs), making the decision to use SLNs in this instance questionable.
  2. We thank the Reviewer for these comments because the issues raised are important as well as at the same time allow us to evidence some positive aspects of our work that are to be taken into account. It should be noted that in this paper is reported an organic solvent free procedure to encapsulate two hydrophilic active principles (DA and GSE) in a lipid matrix and, to the best of our knowledge, is not possible using the standard double emulsification (W1/O/W2) method for encapsulate hydrophilic compounds in SLNs. It is pointed out in Discussion section (pages 15-16 of the manuscript) where we clearly state that our aim is also to evaluate scope and limitations of this our procedure to gain insights into its practical application. We noted that among the limitations of our procedure, there is an increase of the PDI value and we are working in order to solve this drawback and produce SLNs with more homogeneous size distribution and PDI less than 0.3.

As for the comment raised by the Reviewer concerning the low Encapsulation Efficiency of the neurotransmitter (i.e., 62%) it is important to remember that DA is a very potent substance and thereby it is able to produce its effects even at very low concentrations. Overall, on one side, we believe that the co-administration of DA and GSE, rather than one active principle, reciprocally limits the incorporation of the two agents in the lipid matrix of Gelucire® 50/13, so leading to DA's EE of 62%. Indeed, from Figures 5 and 7, the non toxic concentrations of DA/ GSE SLNs fall in the range of 12-25 M of the neurotransmitter which may appear on the low range with respect to the concentrations of unconjugated DA intranasally administered in rats, (namely, 0.3 to 6 mg/mL according to several studies conducted by the group of Maria A. De Souza Silva, using different rat models (Neurobiol Learn Mem 2014, 114, 231-235, doi:10.1016/j.nlm.2014.07.006; Eur Neuropsychopharmacol 2009, 19, 693-701, doi:10.1016/j.euroneuro.2009.02.005.; Neuroscience 2009, 162, 174-183, doi:10.1016/j.neuroscience.2009.04.051)). On the other hand, as a future development of our work, we would like to gain insight into the susceptibility to local nasal metabolism of the neurotransmitter whether it is reduced when DA is carried by SLNs.

About the suggestion to use nanostructured lipid carriers (NLCs) instead of SLNs, we thank the Reviewer for this hint and NLCs investigation could represent for us a useful step-forward of our work, aiming at exploring nano-systems for intranasal administration of DA.

  1. Dopamine, a catecholamine neurotransmitter, oxidizes quickly in water. During production, no anti-oxidants were included.
  2. It is well-known that dopamine, as other catecholamines, undergoes to autoxidation reaction in the presence of air. Some of us, recently, published contributions in this regard [see for instance Trapani et al., Protection of dopamine towards autoxidation reaction by encapsulation into non-coated- or chitosan- or thiolated chitosan-coated-liposomes (2018) Colloids and Surfaces B: Biointerfaces, 170, pp. 11-19; Trapani, et al., Nose-to-brain delivery: A comparative study between carboxymethyl chitosan based conjugates of dopamine (2021) International Journal of Pharmaceutics, 599, art. no. 120453]. However, such autoxidation reaction of the neurotransmitter is slowed down in acidic medium and hence, in such conditions, sample manipulation is possible to same extent. During production of the SLNs herein presented, the acid conditions were established by using dilute acetic (0.01% w/v) as well as an antioxidant agent such as GSE was included (see Section 2.2.1 at page 3 of the manuscript). The mentioned pH dependence of the neurotransmitter autoxidation reaction in solution was previously reported by Umek et al. (Front Mol Neurosci. 11, 467, 2018). In fact, the authors showed that the autoxidation of DA is controlled at acidic pH, while it is very fast in neutral/basic condition as below reported:

pH

K rate (sec-1)

t1/2

5.6

0.000000591

13.5 days

7.1

0.000586

19.7 min

7.4

0.00233

4.95 min

Furthermore, the antioxidant role of GSE exerted towards DA was also observed by us when in “Discussion” Section we wrote (Lines 576-581):  “Of course, when the amount of the antioxidant agent is equal or prevalent in comparison with that of the neurotransmitter, the former may protect DA on the surface of nanocarrier from the spontaneous autoxidation reaction in the presence of molecular oxygen leading to grey-black polymer compounds (e.g., neuromelanins)”. In addition, from Franz cell permeation studies, at Lines 648-652 we also noticed that: “Overall, the presence of GSE represents a benefit for DA permeation because, the prevention of DA autoxidation exerted by this mixture was achieved for both types of SLNs as no color change was seen in the withdrawals during the experiments with Franz diffusion cells, irrespectively of the SLNs tested”.

  1. In the study, the authors did not use translational mucosal models, which are needed for in vitro testing of drug formulations for intranasal administration.
  2. In general, a translational mucosal model should present in vivo-like phenotypic and functional features, such as transepithelial electrical resistance, transmucosal permeation, MUC expression, etc (doi: 10.3390/pharmaceutics11080367). The major problem using ex vivo pig mucosa explants is the limited lifespan of the tissue even under nutritional support. Moreover, they are poor predictors of therapeutic response in humans. To overcome these limitations, translational mucosal models can be used, such as organ cultures, tissues explant cultures, and air-liquid interface (ALI) cultures as well as three-dimensional (3D) cultures (spheroids), obtained with primary airway epithelial cells derived from nasal brushings (doi: 10.1002/lio2.117). Thus, we will focus our forthcoming studies to establish a translational mucosa model for nasal administration.
  3. No in vitro release tests were conducted on the finished product.
  4. We acknowledge the Reviewer for his/her comment. The choice to avoid the performance of in vitro release tests on SLNs was obliged for us because meaningful tests should be carried out at physiological pH 7.4 where a prompt autoxidation reaction occurs, as above displayed. For this reason, in this study we don’t report such tests. Moreover, our “finished products” are represented by hydrogels incorporating SLNs whose characterization will also involve in vitro release tests. Our choice to develop a hydrogel is based also on the possibility that such formulation may further slow down the autoxidation reaction of the neurotransmitter limiting the direct contact of DA with oxygen.

  5. Not enough tests were conducted to support the authors' claims. Large scale testing and characterization are still needed.
  6. High-throughput (HT) screening methods may be useful for large scale testing and characterization of DA delivery via nanostructurated systems, such SLNs or NLCs. Primary nasal cells can be seeded onto permeable insert and upon the generation of an ALI culture, they adopt a pseudostratified columnar compound epithelial layer. Upon full differentiation, the cells produce mucus, display ciliary activity and form tight junctions. Having these permeable support in a 96-well plate may allow HT screening of DA delivery and uptake, as well as monitoring of the mucosal barrier by TEER evaluation (doi: 10.1038/s41598-020-69948-2n). Moreover, these assays can include assessing the irritation potential of nanoformulations or studying how mucus and ciliary action can impact the permeation of a formulation (Henriques et al., Drug Delivery to the Lungs, Volume 32, 2021). Finally, the applications of ALI cultures can be further expanded using co-culture systems. Co-culture with Olfactory Ensheating Cells could be used to assess the nose-to-brain delivery of DA through the intracellular/paracellular pathway. We will investigate these issues in forthcoming experimental set-ups.

Round 2

Reviewer 2 Report (Previous Reviewer 4)

Reviewer finds the manuscript acceptable in the current form.

Author Response

We are grateful to the Reviewer for his/her assessment.

Reviewer 3 Report (New Reviewer)

The authors justified the concerns I had with proper references and explanations. 

Author Response

We are grateful to the Reviewer for his/her assessment.

This manuscript is a resubmission of an earlier submission. The following is a list of the peer review reports and author responses from that submission.

Round 1

Reviewer 1 Report

The authors resubmitted the manuscript and have made changes and additions based on the reviewers' suggestions.

Reviewer 2 Report

Uptake of SLNs by SH-SY5Y cells 48h and 72h experiments has not supplemented. 

Reviewer 3 Report

1.       What method was used to measure the size showed in Table1? Please give the experimental details in Methods part.

2.       Based on the TEM images in Fig S2, the quality of these particles synthesized by this method was not so good which is consistent with the high PDI (from 0.46 to 0.62). The particle (~200 nm) density is so low and the size distribution was wide. How can the authors count more than 200 particles? It seems too hard. Please give some explanations about it.

Reviewer 4 Report

The manuscript deals with the preparation and characterisation of solid lipid nanoparticles as carriers for dopamine and grape seed extract to be applied nasally incorporated into alginate hydrogel. The topic is interesting and well suited to the readership of the journal. However, there are some major and minor issues that need to be addressed by the authors to improve the quality of the manuscript.

More particularly:

1) In the introduction sections authors describe advantages of in situ gelling systems for nasal delivery, being administered as liquid formulations that turn into gel in contact with nasal mucosa. However, in their research authors propose preformed sodium alginate based nasal gel. Such a system could hardly be effectively delivered to to the targeted olfactory region of the nasal cavity as obtainable with in situ gelling systems.

2) In the Introduction section authors stated: „The obtained gels were herein evaluated for in vitro swelling behavior, cytobiocompatibility and cellular uptake.“ In Reviewer's opinion, the main aspect of gel characterisation, i.e. rheological characterisation of SLN loaded and SLN-free gels is needed to get the information on the proposed formulation structure and expected performance. In addition, swelling behavior was analysed with freeze-dried formulations, while cytobiocompatibility/cellular uptake were performed with (as Reviewer calculated from available data) about 6 to 20 fold diluted gel formulation leaving open the question about their performance when applied as a gel in the biorelevant conditions.

3) In the section 2.2.2. GSE adsorbing DA-SLNs, reference „Trapani, A.; Guerra, L.; Corbo, F.; Castellani, S.; Sanna, E.; Capobianco, L.; Monteduro, A.G.; Manno, D.E.; 843 Mandracchia, D.; Di Gioia, S., et al. Cyto/Biocompatibility of Dopamine Combined with the Antioxidant Grape 844 Seed-Derived Polyphenol Compounds in Solid Lipid Nanoparticles. Molecules 2021, 26, 916, should be cited.

4) Page 5, lines 203-2014; „To determine DA and GSE content in both labelled and unlabeled SLNs, freeze dried particles were cleaved upon enzymatic digestion by esterases.“ – the details on freeze-drying are missing.

5) 2.6. Assessment of physical stability of SLNs – apart from particle size, no other stability parameters were evaluated, like zeta-potential or drug content in the formulation, although they are crucial to derive the conclusion on SLN stability profile. In addition, studies dealing with the stability of SLN loaded alginate hydrogel that present the proposed form to be applied nasally, should be performed as well to assess the feasibility of the formulation approach proposed.

6) The swelling studies should be better explained. Firstly, authors should clarify why freeze-dried gels were used in swelling studies, instead of hydrated form that is intended to be applied nasally. In addition, it is not clear if the freeze-dried formulation was immersed in SNF or brought in contact with SNF (e.g. over the semipermeable membrane), which would better simulate conditions in vivo. Also, it is not clear how the resulting gel was dried in the end of procedure: „By discarding supernatant, the resulting gel was accurately dried and subjected again to weight determination.“ If it is completely dried, than SNF content cannot be derived from the weight difference. If not, the Reviewer find it confusing that the final weight was 15 times lower than starting, as stated (lines 480-483; Precisely, Na Alg gels incorporating DA-co-GSE-SLNs provided the highest amount of shrinking probably due to the SNF loss, being the original (dry) weight 15 times higher with respect to the weight of the formulation kept in SNF for 24h (weight at time 0 vs weight at 24h, p≤0.001).“Authors should also comment on the experimental setting of the temperature (37 degrees) and duration of swelling studies (24 h), taking into account the physiological parameters and expected retention time of the formulation at the nasal mucosa. The same refers also to permeation studies.

7) Ex vivo SLN permeation studies; Page 7, lines 304-307; „For the experiments, the anterior surface of the mucosa was faced towards the donor compartment (effective diffusion area = 0.6 cm2), and freeze-dried DA-co-GSE-SLNs (or GSE-ads-DA-SLNs) were resuspended in 100 uL of SNF (pH 5-6) and placed in the donor compartment.“ It is not clear why only SLN formulations were used for permeation studies, and not the gel encorporating SLN. In addition, why freeze-dried SLN formulations were used – if so, particle size and zeta potential should be determined upon resuspension and before permeation studies. For example, SLN agglomeration owing to freeze-drying could reduce the DA flux across the barrier. In addition, authors stated: „Each withdrawal was subjected to centrifugation (Eppendorf 5415D, Hamburg, Germany) at 13,200 x g, 45 min, to allow discarding of the pellet.“ The samples are withdrawn from the receptor medium. It is not clear why pellets could be found in such samples? Did the authors check the recovery by adding together the DA amount found in the receptor medium, within the mucosa, and the fraction retained in the donor compartment in the end of experiment?

8) Line 329 – the growth medium should be specified.

9) Lines 330-331; „OECs were then incubated with 0.01 % of Na Alg gel embedding 12 µg/mL of FITC in the SLNs and 0.003% of Na Alg gel embedding 4 µg/mL of FITC in the SLNs for 24h.“ – authors should specify the meaning of the numbers 0.01% and 0.003% where first mentioned, as well as define the freshly prepared gel composition in the same manner. Authors should also clearly state the factor of dilution of the freshly prepared gels when preparing samples to be tested on porcine mucosa and cells.

10) 2.11. Evaluation of SLN internalization in OECs; in the internalization study FITC solution should have been used as control to enable evaluation of influence of SLN on FITC internalisation. The same comment refers to 2.13. Evaluation of uptake by SH-SY5Ycells.

11) Lines 341-342; „Precisely, 0.01% of Na Alg gel embedding 12 µg/mL of FITC in the SLNs and 0.003% of Na Alg gel embedding 4 µg/mL of FITC in the SLNs were the tested dilutions.“ Are these concentrations relevant for deriving conclusions of gel influence of SLN internalisation, taking into account the multiple dilution, and probbable loss of the gel structure (it can be inspected by rheological studies)?

12) Lines 406-407; „Regarding active principle content“ – what does this expression stand for?

13) Permeability studies should be performed with SLN-loaded gel formulations in addition to SLN formulations.

14) Line 594; „Acetic acid DA“ – what does this expression stand for?

15) English should be revised. There are parts of the text that are difficult to follow, e.g. in abstract: „Fluorescent SLNs were prepared and, at concentration of 0.01 % and 0.003% of Na Alg gel, embedding 12 µg/mL and 4 µg/mL of FITC loaded by SLNs, no citototoxicity was evidenced in Olfactory Ensheathing cells (OECs). At these concentrations of fluorescent SLNs, once incorporated in sodium alginate gels, then in vitro they were observed to be taken up within 24h by OECs.“